# Gambling and Aging: An Overview of a Risky Behavior

**DOI:** 10.3390/bs13060437

**Published:** 2023-05-23

**Authors:** Maylis Fontaine, Céline Lemercier, Céline Bonnaire, Isabelle Giroux, Jacques Py, Isabelle Varescon, Valérie Le Floch

**Affiliations:** 1Cognition Lanque Langage Ergonomie, Centre National de la Recherche Scientifique, Université Tou-louse-II-Jean-Jaurès, CEDEX 09, 31058 Toulouse, France; celine.lemercier@univ-tlse2.fr (C.L.); jacques.py@univ-tlse2.fr (J.P.); valerie.le-floch@univ-tlse2.fr (V.L.F.); 2Laboratoire de Psychopathologie et Processus de Santé, Université Paris Cité, 92100 Boulogne-Billancourt, France; celine.bonnaire@u-paris.fr (C.B.); isabelle.varescon@u-paris.fr (I.V.); 3Centre Québécois d’Excellence Pour le Traitement du Jeu, Université Laval, Quebec, QC G1V 0A6, Canada; isabelle.giroux@psy.ulaval.ca

**Keywords:** aging, older adults, gambling, gambling disorder, risk-taking, behavioral research

## Abstract

Gambling is a field of study that has grown since the 2000s. Much research has focused on adolescents and youth as a vulnerable population. The rate of aging gamblers is increasing; however, evidence-based knowledge of this population is still too sparse. After introducing the issue (1), this article provides a narrative review of older adults’ gambling through three sections: (2) older adult gamblers (age, characteristics, and motivations), (3) gambling as a risky decision-making situation, and (4) gambling disorder related to older adults. By drawing on the existing literature from a problematization perspective, this type of review can highlight complex and original research topics and provoke thought and controversy to generate avenues for future research. This narrative review provides an overview of the existing literature on gambling among older adults and offers perspectives on how aging can affect decision-making and thus gambling for this population. Older adults are a specific population, not only in terms of the consequences of gambling disorders but also in terms of the motivations and cognitions underlying gambling behaviors. Studies on behavioral science focusing on decision-making in older adults could help in the development of public policy in terms of targeted prevention.

## 1. Introduction

Gambling refers to any operation offered to the public, under any name whatsoever, in order to raise the hope of a gain which is due, even partially, to chance, and for which a financial sacrifice is required from the participants [1]. Gambling is therefore an activity that relies on chance and in which someone risks money in order to win. This risky activity is increasingly part of the daily lives of people from many cultures [2,3,4] and is of concern to the World Health Organization itself [5]. From North America to Australia, Europe, Africa, and Asia, more than one in two individuals on average report having gambled in the past year [6,7,8,9,10]. Among older adults from different countries, those with a non-Caucasian cultural identity range from 26.6 to 56.2 percent to gamble [11], with participation across all cultures ranging up to 85.6 percent [12]. In France, 49.3% of 55–64 year olds and 37.6% of 65–75 year olds gambled in 2019 [7]. Older adults have positive attitudes towards gambling, which is seen as an ordinary and safe form of entertainment [11,13]. While 79.5% of people over 60 say they gamble for entertainment, 62.7% also say they gamble to win money [14]. Additionally, it is when gambling is perceived as a reward in itself that gambling is risky [15].

How can the gambling attractiveness among older adults be explained? The occupational, emotional, financial, and health-related changes experienced by this population mean that they have a lot of free time available and a different relationship with money, leisure, and the future, making older adults a prime clientele for gambling promoters [15]. In a Silver Economy logic [16], older adults are the target of many marketing campaigns [17,18], which present gambling offers as the perfect answer to seniors’ needs (organized service road to casinos, first bet offered, and promise of socialization), as well as a means to fight isolation [19]. Isolation is the common theme in the three trajectories that can lead older adults from recreational to gambling disorder: a habit pathway associated with gambling habituation, a dormant pathway marked by pre-existing excessive behaviors or impulsivity, and a grief pathway associated with unresolved losses [20]. Increased availability and exposure to gambling products and establishments can send older adults down the habit pathway. Since increased exposure leads to gambling disorder, rates of gambling disorder among older adults are likely to increase [21]. Numerous studies have shown an increase in participation rates and rates of gambling-related disorders among older adults (for a review, [22]). These increases are expected to become even more pronounced in the coming years simply because of the population aging [23].

Although the prevalence rates of gambling disorder are lower among older adults than among their younger counterparts, it is a problem of concern among this population [11,24], particularly because of the severity of the consequences, such as financial loss, debt accumulation, and illegal behaviors (forging, fraud, theft, or embezzlement to finance gambling) directly related to gambling [23]. Fixed incomes, in particular, make older adults an extremely vulnerable group [24]. However, the issue of aging in gambling is still too rarely investigated. Longitudinal studies and collaborative research combining epidemiology, sociology, psychology, and behavioral analysis are lacking, and the current work is mostly descriptive and exploratory [11]. The literature also lacks empirical data [12,25]. While there are 13,423 publications archived on Psychinfo from 1973 to 2022 with the keyword “gambling” alone, the keywords “gambling and aging” or “gambling and elderly” only return 258 and 86 publications, respectively. If we add the keyword “experiment”, the number of publications drops to eight and zero, respectively. Cross-linking the literature on older adults’ gambling with that on decision-making in risky situations offers, however, some avenues for behavioral research.

## 2. Older Adult Gamblers: Age, Characteristics, and Motivations to Gamble

### 2.1. Older Adult Gamblers’ Age

Although the World Health Organization set the age of entry into old age at between 60 and 65, this institution recognizes that old age must be defined by new roles and various factors other than biological age [26]. Even though the term “senior”, which appeared in the early 1990s, refers to those over 55 years of age or even over 45 years of age in the workplace [27], some people have proposed the probability of surviving 10 years as an indicator of entry into old age, thus advancing the age of old age in line with the increase in demographic aging [28]. The age of old age is therefore a matter of debate. There is also no consensus in the gambling literature, with the lower bound being, depending on the study, 50 years (e.g., [29,30]), 55 years (e.g., [2,12,17,31,32,33,34,35,36,37]), 56 years (e.g., [38]), 60 years (e.g., [14,39,40]), 65 years (e.g., [41,42,43]), or sometimes 70 years (e.g., [44]).

The chosen limit is only rarely justified. For Asian studies, the reason is that retirement is at age 55. Granero et al. [30] justified the lower limit of 50 years because of the large variations in the literature. Pilver et al. [35] defined the limit of 55 for comparability with previous work. In Connecticut, Petry [45] found that older female gamblers began regular gambling at age 55, whereas regular gambling among older male gamblers began before age 30. The large prevalence survey on substance use and gambling among seniors [46] in New Brunswick, Canada, set the cut-off at 55. Lastly, age 55 was identified as a reasonable threshold for the onset of age-related impairment in decision-making [47].

### 2.2. Older Adult Gamblers’ Characteristics

The question of the profile of the older gambler does not lead to any definitive typical conclusion. There are variations by gender and socioeconomic level in the general population, but there is no clear pattern among older adults [24]. Some studies suggest that the prevalence of gambling disorder among older adults is higher among men than women [48,49], but others indicate that it is similar or even higher among women [50,51,52,53]. At the socio-economic level, while education [49] and income [54] have been identified by some studies as risk factors, others have found no link [48]. Regarding types of gambling, there are games of pure chance, non-strategic, such as lotto or casino games, and games involving a degree of skill such as poker, sports betting, or horseracing [55]. Older adults prefer non-strategic games over strategic games [22,56], and the prevalence of gambling disorder is greater in casino games such as slot machines than in lotteries [53]. The theoretical model developed by Tira et al. [20] proposes three trajectories to gambling disorder. The habit pathway is characterized by repeated exposure to a gambling practice and conditions (“beginner’s luck” and big wins) that foster the entrenchment of erroneous beliefs (gambler’s fallacy, superstitions, and near-miss), which lead to an escalation of gambling activity. From a reason to start gambling, breaking the boredom becomes an excuse to gamble more. The dormant pathway relates to predispositions to compulsive behaviors. This pre-existing vulnerability, along with strong co-morbidities (tobacco and alcohol dependence) and low levels of self-control, leads to significant risk-taking for the sheer pleasure it provides. In contrast to this presumed biological or genetic predisposition of the dormant pathway, the occurrence of certain negative life events (essentially losses of various kinds and intensities) characterize the grief pathway. The emotional vulnerability induced by losses (of role, of loved ones, of income, or of physical abilities) gives rise to a need to gamble and to respond to the incentives linked to gambling, in order to escape negative emotions such as loneliness, despair, depression, or anger [20]. For example, older adults are likely to increase their gambling activity in the hopes of offsetting the loss of income [53]. Motivation to win money, combined with excitement seeking, predicts gambling disorder, especially among slot machine gamblers over 60 [53].

### 2.3. Older Adults’ Motivations for Gambling: Risk Factors or Not?

Of the studies that have examined gambling in aging people, the largest number have focused on motivations, with no universal consensus. While New Zealand older adults gamble to win money more frequently than younger gamblers [52], this is clearly not a pronounced motivating factor among British older adults [21]. For the latter population, gambling is an escape from the psychological (loneliness or grief) and physical (pain and decline) stress and discomfort associated with aging. The high accessibility and availability of gambling environments compared to other leisure activities make it a warm and feel-good activity [21]. The most common reasons reported by casino gamblers over 65 are to relax (68.1% of respondents), relieve boredom (30%), pass the time (42.7%), and go away for the day (43.7%) [57]. Of the five dimensions of motivation to gamble for older Americans (i.e., thrill of winning, escape, socialization, fun, and curiosity), the primary motivation is fun and pleasure [58]. From the perspective of self-determination theory [59,60], the main motivations of casino gamblers over the age of 60 refer to intrinsic motivations (i.e., the activity is a reward in itself), such as entertainment and pleasure. Two-thirds of respondents, however, expressed extrinsic motivations (i.e., the activity has an instrumental value or a utilitarian purpose), such as financial gain. More self-determined reasons for gambling (intrinsic motivations) are associated with greater participation in gambling, and less self-determined reasons (extrinsic motivations) are associated with less participation in gambling [61]. Beyond simple participation, for both older adults (M_age_ = 74.59) and younger adults (M_age_ = 22.61), the severity of gambling disorder is more associated with a tension-relief motivation than with money-making and thrill-seeking [52]. Motivational factors related to psychological and physical stress reduction [21] are similar, in the general population, to the coping motivations, via negative reinforcement, of the “behaviorally conditioned gambler” [62]. This subgroup of problem gamblers has no premorbid characteristics but has comorbidities that are consequences, not causes, of repeated problem gambling behaviors. These gamblers fluctuate between regular and problem gambling due to the effects of conditioning, increased vulnerability to erroneous cognitions, errors in judgment, and poor decision-making. With the least severe difficulties of all the problem gamblers, this subgroup is motivated to enter treatment and can successfully return to controlled levels of gambling after treatment [62].

Another motivation for older adults’ gambling is a desire to exercise one’s mind [12]. Gambling can, indeed, satisfy unmet psychological needs for not only emotional but also cognitive stimulation [21]. One of the reasons for the popularity of gambling among older adults is that it provides an opportunity for cognitive stimulation [63,64,65], which is thought to improve cognitive functioning and thus counteract the well-documented decline in cognitive performance associated with aging [47,66,67,68,69]. Recreational gambling among older adults is associated with positive outcomes in terms of cognitive stimulation, socialization, health, and well-being [42,70,71]. In a leisure state, gambling increases activity, develops socialization processes, and stimulates cognitive functions, thus favoring better overall health [72]. Recreational gamblers over 65 report higher levels of “good to excellent” general health than non-gamblers [42]. Gamblers over the age of 60 develop cognitive and behavioral strategies to prevent them from falling into gambling disorder [71] and to keep their gambling under control. On the cognitive level, this involves keeping in mind the real dangers of gambling, being aware that the odds are stacked against one, and never considering oneself lucky; on the behavioral level, it involves regulating one’s practice through concrete actions such as only taking a predefined amount of money, not gambling alone, stopping and not reinvesting one’s winnings, and optimizing one’s gambling time by betting small sums. Thus, recreational gambling has a number of beneficial effects. On the other hand, problem-gambling behaviors have been associated with high levels of anxiety and loneliness [73,74]. Older adults may perceive gambling as a way to reduce negative mood states [43]. The need for protection and assertiveness contributes to sustained gambling [75]. When the primary motivation for gambling is to escape anxiety and loneliness, the likelihood of experiencing gambling-related harms is greater [23,37,43].

## 3. Gambling: A Risky Decision-Making Situation in Which Older Adults Are Likely to Be Vulnerable Due to Their Specificities

### 3.1. Decision-Making, Risk-Taking, and Gambling

Gambling is an activity whose outcome is based mainly or entirely on chance and involves an irreversible prior stake of money or something of value [76]. As defined by Rogers [77], gambling is a situation in which the gambler decides to take the risk of losing, with the hope of winning according to some probabilities. The cognitive approach distinguishes three types of decision-making situations [78]: under certainty (decisions for which we know and are absolutely certain of the outcome), under uncertainty (decisions for which we do not know the probability of the outcome), or at risk (decisions for which the outcomes are known as well as the probabilities of these outcomes). A risky decision is, for example, a situation in which there is a 30% chance of doubling one’s stake [79]. In such a situation, we know the possible outcome of the stake (doubling it) and the probability of this outcome (about one chance in three). However, although we know the probability of this consequence, we cannot be certain of the occurrence of the desired outcome (doubling the bet), so there is still some uncertainty. The neuroscientific approach distinguishes between decision making under ambiguity (the probability of obtaining a specific outcome is unknown or is a matter of chance, the possible choice—safe versus risky—does not differ in terms of the value of the reward and there is no greater risk in making either choice) and decision making under risk (the probability of each outcome is known and the participant has to decide between a safe choice—associated with a high probability of obtaining a relatively small reward in value—and a risky choice—associated with a low probability of obtaining a significantly larger reward in value). From an ecological perspective, the gambling situation involves making a decision under both uncertainty (Kahneman and Tversky [80,81] have shown that humans have great difficulty processing probabilities) and risk (known or potentially known outcomes and probabilities).

Although widely investigated in the decision-making literature developed from economic theories [82], risk-taking has been studied in only a few gambling studies. Decision-making is a complex cognitive process aiming at the selection of a type of action among different alternatives in order to lead to supposedly rational decisions and to the formation of judgments [83]. Indeed, the decision-making process involves “specifying possible courses of action, collecting information about the alternatives, identifying likely future events and other circumstances relevant to the decision, and considering the possible outcomes contingent on the chosen action and the prevailing circumstances” [84] (p. 297). Identified as early as the 1950s, the discrepancies between the normative prescriptions of economic theory (assuming rationality in decision-making) and real human behavior are at the origin of the heuristics and biases trend and Kahneman and Tversky’s studies [83]. These authors’ theory postulates that the reason why the treatment of information about probabilities and consequences escapes normative economic theory is because it is governed by perceptual and attentional mechanisms. It provides a richer and more ecological behavioral model to better account for the complexity of human decision-making in real life than traditional economic theory [82].

More precisely, risk decision-making occurs when individuals make choices whose consequences depend on the probability of occurrence of certain events [85]. The prospect theory [86] describes the mental process of decision in two phases; it assumes that risk-taking behaviors vary according to the definition of the risky situation. The first phase, called editing, where the individual subjectively represents the perspectives offered in a decision problem, depends on the type of information processing operated. Then, comes a phase of evaluation of the previously edited perspectives, which allows a choice to be made based on the highest subjective utility. The calculation of this subjective utility integrates two functions [85]: a function of the subjective value relative to a reference point and a function of probability weights relative to the coding of perspectives in a domain and the type of probability (see Table 1).

Figure 1 illustrates how loss-zone functions reflect both risk-taking behavior and loss aversion. Because the annoyance of losing a certain amount of money is greater than the pleasure of winning the same amount [86], the subjective value function in the loss domain (on the left of the vertical axis) is not symmetrical to that in the gain domain (on the right), relative to a reference point (represented by the horizontal axis). In fact, the gains curve rises less quickly than the losses curve, which means that the gains become less valuable than the losses as one moves away from the reference point.

With reference to prospect theory, the inclusion of the gambler’s state of wealth at the time of betting as a reference point for the definition of the gambling situation makes it possible to calculate the risk-taking of participants [87]. This is especially interesting because unlike the research on gambling, the research on decision-making has massively examined the effects of age (for a multi-disciplinary review about aging, see [88]). Decision-making changes with aging [82]. Older adults tend to make decisions that are less advantageous compared to younger adults [47,68].

### 3.2. Older Adults’ Risk-Taking and Decision-Making

Mather et al. [89] studied the risk-taking of adults aged 55–89 with respect to the certainty effect (see Table 2) defined by Kahneman and Tversky [86] whereby, in addition to the desirability of the outcome (gains being desirable and losses undesirable), individuals perceive certain outcomes (such as a strictly 0% or 100% chance of the outcome occurring), as different from and more influential in decision-making than uncertain outcomes (all other probabilities between 0 and 100%).

The certainty effect thus predicts risk aversion when it comes to gains and risk seeking when it comes to losses. Older adults are more sensitive than younger ones to this certainty effect [89]. Older participants take more risks than younger ones when it comes to potential losses but also fewer risks than younger ones when it comes to potential gains [90,91]. It thus appears that attitudes towards risk change with aging. Moreover, the decisions of participants aged 65 to 90 years lead to the lowest monetary results: 39% less earnings than young people aged 21 to 25 years and 37% less earnings than adults aged 30 to 50 years [91]. Finally, despite a high IQ (above 110) guaranteeing appropriate calculation and reasoning faculties and a very satisfactory Mini Mental State Examination (MMSE) score (M = 29.03/30, SD = 0.9), attesting to the good mental capacities of the participants, the majority of the elderly showed striking and costly inconsistencies in their choice behaviors between the different monetary proposals [91]. The oldest adults thus show an impairment of decision-making despite their intellectual faculties. The normal process of cognitive decline leads, in certain situations, to deficits in decision-making after age 60 [66]. Some authors consider these deficits to suggest the possibility that some apparently normal older people show disproportionate aging of the ventromedial prefrontal cortices [68], but there is no clear, consensus explanation for the impairment of decision-making with age. The fact is that adults between the ages of 56 and 85 show a poorer performance on the Iowa Gambling Task (IGT, decision-making task under ambiguity where the participant must draw cards from four piles and maximize his or her winnings) than younger adults [47]. These impairments in decision-making also form the basis of gambling disorder.

### 3.3. From Decision-Making to Gambling Disorder

Gambling disorder is considered a mental disorder that impairs decision-making in risky situations [92]. Looking at risk-taking from the gambling disorder point of view, it reveals that loss aversion is unevenly distributed among gamblers with the disorder. The 57 subjects (mean age 34.1 years) in Takeuchi et al. [92] were asked to choose between two options, each of which offered a 50% chance of a certain alternative occurring versus a 50% chance of the other (see Table 3).

The distributions of loss aversion were significantly different: the control group (*n* = 26) showed a relatively homogeneous level of loss aversion (12 subjects with low aversion, 7 with medium aversion, and 7 with high aversion), whereas the gamblers with a gambling disorder (*n* = 31) showed heterogeneous loss aversion (19 subjects with low aversion, 1 with medium aversion, and 11 with high aversion). This suggest that gambling disorder is thus not be a uniform disorder but a heterogeneous one, with a marked and polarized difference in attitudes to risk. Combined with their increased participation in JHAs [93], the greater loss aversion among older adults [89] may be an additional factor in the growth in gambling disorder prevalence expected in this population as a result of demographic aging [30]. Although some adaptive processes, such as the use of an intuitive thinking system, ensure that older adults make satisfactory decisions in some contexts [94], the specific context of gambling may work against their efficiency. While intuitive decision making saves cognitive effort, the associated heuristics may favor systematic biases [95]. In a gambling situation, the use of heuristics is a harmful bias, leading to irrational decision-making and risk-taking where emotion plays a definite role [96]. The age effect variations depending on heuristic types also appear to rely on factors beyond cognitive aging, such as motivation and emotion [88]. When making a decision, older adults focus more than younger ones on emotional content and sometimes even on specifically positive content [82].

## 4. Gambling Disorder: How Is It Related to Older Adults through Control?

Additionally known as excessive gambling, problem gambling, gambling addiction, or pathological gambling, gambling disorder is a DSM-5 term referring to a chronic and progressive illness characterized by a preoccupation with gambling and a need to increase the value and frequency of bets to achieve a desired level of excitement [97]. Gambling disorder include but are not limited to compulsive gambling and refer to all gambling behaviors that may disrupt or interfere with family, personal, or professional activities [98]. The concept of use disorder bases the diagnosis on the use mode of an object (substance or not). The use of the object, which is a source of pleasure, is renewed despite its harmful repercussions in various areas of daily life. This puts a strong hypothesis of the cause of disorders linked to use, which is located at the level of a disorder of the control system [99]. Impaired activity in the prefrontal cortex has been identified in the decreased cognitive control over gambling urges [100]. The role of the prefrontal cortex in the reward system is thought to be regulated by dopaminergic transmission, which is dysfunctional in individuals with gambling disorder [101]. The hypothesis of an alteration in dopaminergic pathways has also been put forward for behavioral addictions (gambling, compulsive purchases, sexuality, and eating) linked to Parkinson’s disease treatments [102]. According to Guillou Landreat and colleagues [53], this should be put in perspective with the altered regulatory relationship between dopamine and prefrontal reward-related activity detected in healthy older adults [103]. These alterations may reduce older adults’ ability to control their gambling activity. Beyond brain mechanisms, this type of control is also a matter of an individual’s belief about whether or not they can resist an opportunity to gamble in a given situation [104], with the object of control being the behavior. This belief refers to the self-efficacy notion, defined as a system of beliefs about one’s own self-efficacy and one’s ability to organize and execute the actions required to produce given results, which influences performance through its motivational function [105]. Since aging is accompanied by losses that reduce control over the environment, situations of deprivation of a sense of control are assumed to increase with age. They are thus likely to develop in older people a belief in learned helplessness, defined as a generalized abandonment behavior resulting from the experience of uncontrollable aversive events [106]. Maintaining a sense of self-efficacy despite diminished abilities is beneficial adaptive functioning for older adults [107]. Individual performance is better predicted by beliefs about abilities than by actual abilities [108]. The illusion of control may thus make it possible to limit the feeling of acquired powerlessness, caused by repeated confrontations with situations of deprivation of the sense of control [109] and may thus generate benefits for older people [110]. While control (even illusory) over gambling behavior (see, e.g., Casey et al. [104] for work on self-efficacy), guarantees risk-free practice, the illusion of control over the game itself, as a tendency to overestimate the weight of one’s actions in situations determined, in reality, by chance (a factor beyond one’s control), is, on the contrary, a well-known risk factor [111,112]. We are therefore faced with a paradox where, in order to “gamble responsibly”, the gambler is encouraged to show control, but only over his or her behavior, while remaining aware that he or she has no control over the game [113]. Since reducing gambling-related cognitive distortions (such as the illusion of control) is one of the best predictors of recovery from gambling disorder [114], cognitive distortions in older adults merit investigation for targeted prevention [53]. Their identification in the perception of gambling by Asians over 60 reveals that three levels of cognitive distortions play a role in the maintenance and escalation of their gambling practice [115]: these erroneous cognitions relate to probabilistic control (chasing wins or hot hands, trying to recover from cold hands, and gambler’s fallacy), interpretive control (the belief that there are more wins than losses), and illusion of control (perception of competence, near wins, luck, and superstitions). However, older French people (55+) show less illusion of control than very young people (18–25), even though they take more risks in simulated gambling [116]. It should be noted that, in addition to the cultural difference, the majority of the French participants were non-gamblers or occasional gamblers, whereas the Asian participants were regular gamblers.

The consequences of gambling disorder are of concern, with older gamblers (65+) having greater difficulty recovering from health complications, psychological and social problems, and financial difficulties resulting from problem gambling [117]. Indeed, older adults with gambling disorder, who are less likely to be identified and less likely to seek help, appear to be more vulnerable (financially fragile and at risk of suicide) than other age groups [118]. It should be noted that some gamblers, on the contrary, combine old age with wisdom by moving from gambling disorder to recreational gambling as they age [15]. Although rates of participation and gambling disorder tend to decline with age, they increased for both younger and older people between 1997 and 2000 [93]. More research is needed on the gambling behaviors of older gamblers and their determinants in order to identify the tipping point from leisure to addiction.

## 5. Discussion

Generally perceived by older adults as a safe leisure activity, gambling is, for them, a way to break from isolation and boredom [11,12]. While the motivations of aging gamblers may vary across cultures, entertainment and pleasure are universal motivations [14,58]. Recreational gambling among older adults is even associated with positive outcomes in terms of socialization, health, well-being, and cognitive stimulation [42,71]. In contrast, when the main motivation is to escape anxiety and loneliness [73,74], the consequences are harmful [30,37,43] and more deleterious in older adults than in younger ones [117,118]. In addition, age-related changes in decision-making processes are a vulnerability factor for gambling disorder. They lead older adults to take greater risks in uncertain contexts, especially those involving losses [88,89,90,91].

The older adults’ isolation is not only a risk factor for problem gambling in this population, but it can also interfere with the effectiveness of therapies [119]. Family or social support is essential to therapeutic effectiveness [120]. Although studies lack standardized methods to draw solid conclusions, cognitive-behavioral therapy has shown some effectiveness on gambling disorders. The effects are not immediate, but improvements in gamblers’ quality of life and decreases in stress, anxiety, depression, gambling frequency, and wagering amounts are seen at three, six, and twelve months [121]. Interventions that use a manual and those that combine cognitive-behavioral therapy with mindfulness or Gamblers Anonymous support group sessions are the most effective. Group cognitive-behavioral therapy appears to be an effective treatment approach for both restoring self-control over gambling behavior and reducing the emotional distress associated with this disorder [122]. Research on the effectiveness of cognitive-behavioral therapy in the elderly has highlighted its major value in older adults, who benefit from it to a degree roughly equivalent to that of younger adults. Although the cognitive-behavioral theoretical foundations remain the same, specific adaptations of therapeutic strategies and processes can optimize results in the elderly [123]. Care must be taken in group therapy to ensure that all group members have similar cognitive abilities; otherwise, the progress of all participants may be compromised. Realistic and achievable objectives, adapted to the patient, must be considered [120]. The use of a specific format that focuses more on behavioral interventions is sometimes advocated [124].

## 6. Conclusions

Before concluding, we note the limitations of this narrative review. The methodological criteria are lower than in a systematic review, which may include a snowballing literature search [125]. However, this type of review allows us to have a general view of a specific subject, to raise issues that are usually neglected, and to encourage further research on this subject [126,127,128].

Older adults are a specific population for gambling issues, not only in terms of the consequences of gambling disorder [98,117,118] but also in terms of the cognitions underlying gambling behaviors [116]. To understand the mechanisms that cause older adults to shift from leisure to addiction, we need to understand the specific behaviors of these aging gamblers. While the value of behavioral approaches to understanding and preventing substance use disorders is clear [129], there is also much work to be conducted on non-substance use disorders such as gambling, and particularly with populations whose specificities need to be examined, such as older gamblers.

## Figures and Tables

**Figure 1 behavsci-13-00437-f001:**
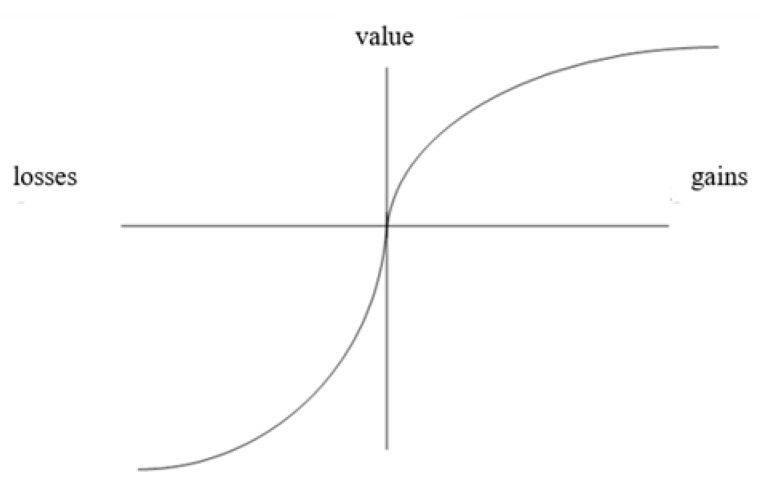
The value function [85,86].

**Table 1 behavsci-13-00437-t001:** The four attitudes towards risk [85].

Coding Domain	Type of Probability	
	Low Probabilities	Medium and High Probabilities
Gains	Risk seeking (e.g., preference for a bet with a 1% chance of winning USD 5000 over a sure gain of USD 5)	Risk aversion (e.g., preference for a sure gain of USD 500 over a bet with a 50% chance of winning USD 1000)
Losses	Risk aversion (e.g., preference for a sure loss of USD 5 over a bet with a 1% chance of losing USD 5000)	Risk seeking (e.g., preference for a bet with a 50% chance of losing USD 1000 over a sure loss of USD 500)

**Table 2 behavsci-13-00437-t002:** Choice under the certainty effect.

Domain	Outcome Probability of Occurrence	
	0% or 100% (Sure Outcome)	Between ]0 and 100[% (Risky Outcome)
Gains	Preferred choice	Avoided choice
Losses	Avoided choice	Preferred choice

**Table 3 behavsci-13-00437-t003:** Material from Takeuchi et al. [92].

Options Available to Participants					
Option 1					
50% chance of losing a fixed amount X (2500, 5000, 10,000, or 15,000 yen depending on the trial)			50% chance of winning an amount Y (between 0.5×X and 10×X)		
Option 2					
50% chance of losing 0 yen			50% chance of winning 0 yen		
Loss aversion parameter (λ) calculated from the prospect theory of Kahneman and Tversky					
Low aversion		Medium aversion		High aversion	
0 < λ < 3.33		3.34 < λ < 6.66		6.67 < λ < 10	

## Data Availability

No new data were created or analyzed in this study. Data sharing is not applicable to this article.

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
