# Peer review of "Gambling and Aging: An Overview of a Risky Behavior"

_behavsci, 2023, doi:10.3390/bs13060437_

Round 1

Reviewer 1 Report

Dear Authors,

You have presented an interesting discussion of the literature on older people's gambling, with attention to older gamblers' characteristics and motivations, decision-making and gambling disorder. 

My comments and suggestions for improvements are in comments in the attached PDF. 

Overall, it is perfectly acceptable as it is. I feel that it could be improved with a slightly more critical eye in parts, and/or a couple more suggestions for future research. 

All the best.

Author Response

We thank the reviewer for her/his comments that helped us improve the manuscript. We have responded to each of the comments made on the PDF file. Please see the attachment.

Reviewer 2 Report

The submitted article is in line with the focus of the journal and the topic is interesting. The overview of the discussion on "gambling and aging" is successful and clear.

However, I would still recommend some improvements:

·       Quite often, there are several quotations to one idea. In my experience, the authors of the articles/books do not always say the same thing, more discussion is appropriate, e.g. line 32 [quotes 5-7], line 35 [quotes 9-13], line 385 [quotes 92-95, 123,124].

·       Please check that the sentence starting on line 35 is correct. Is it correct non-Caucasian cultural identity or Caucasian cultural identity? The sentence goes on and on and as a whole, it does not make much sense.

·       Please check that the sentence starting on line 71 is correct, especially “publications drops to 8 and 0 respectively”.

·       In Figure 1, I would mark the places where there is an inclination or aversion to risk, etc., as described in the text.

·       The source for Table 2 is missing.

·       The article would certainly benefit from more discussion of the socio-demographic factors affecting gambling among the elderly.

·       There are a number of typographical errors:

·       In a sentence starting on line 84, the age is always given followed by (e.g., [source]). Sometimes, however, parts of the parentheses are missing. Missing ")" after [41] and after [44-46].

·       The quotation marks are wrong on line 109. Now it's <" big wins> and it should be <"big wins">.

·       Lines 144 and 145 probably give the "mean age" in the abbreviation “Mage=”. It would be useful, for example, to put "age" in subscript or italics. This way it looks like the word "mage".

·       On line 213, consider putting the page citation inside the source citation, i.e. instead of “[87] (p. 297)” write “[87, p. 297]”.

·       The last column in Table 2 is "Between ]0 and 100[". Why are there square brackets and vice versa? It makes sense without them.

·       Unify upper and lower case for the word "choice" in Table 2.

·       I would reformat Table 3 to make it narrower. It does not make sense why it should overlap.

I recommend proofreading and formatting before final submission.

Author Response

We thank the reviewer for her/his comments that helped us improve the manuscript. We have responded to each of them. Please see the attachment where the reviewer's comments are in italics, each followed by our response.

Reviewer 3 Report

The present article deals with a very interesting topic. The article is well conceived and I appreciate the way the authors have approached the article. The theoretical background is a good reflection of the article itself. The authors have used a number of relevant and up-to-date sources. Chapter 1 is logically structured from general information to specific ones. It introduces the reader to the issue of gambling and risk phenomena in the aging population. It is good that the authors do not directly specify old age and aging, but the primary issue is pathological behavior. They follow this up with other relevant information that provides a comprehensive overview of the issue. I would make only partial changes to make the article fully publishable.

I would recommend changing Chapter 5 to a discussion and adding more general information that may be applicable to this pathological behavior. It changes quite often about multidisciplinary approaches that can be applied to the elderly. Some information on this issue would be useful here.  (E.g. https://www.webofscience.com/wos/woscc/full-record/WOS:000618092500006 ; https://www.webofscience.com/wos/woscc/full-record/WOS:000187345200002).

Next, I would create a chapter 6, which would be defined as a conclusion. This chapter will then come to some summary and general recommendations for practice from the authors' perspective. It is also useful to mention any limitations of the study and a forecast of the issue.

In case of such a modification, then the article is definitely recommended to be published.

Author Response

(The authors gave the same response as above.)
